# Do Serum C-Reactive Protein Trends Predict Treatment Outcome in Patients with Knee Periprosthetic Joint Infection Undergoing Two-Stage Exchange Arthroplasty?

**DOI:** 10.3390/diagnostics12051030

**Published:** 2022-04-20

**Authors:** Sheng-Hsun Lee, Chun-Ting Chu, Chih-Hsiang Chang, Chih-Chien Hu, Szu-Yuan Chen, Tung-Wu Lu, Yu-Chih Lin

**Affiliations:** 1Department of Biomedical Engineering, National Taiwan University, No. 1, Sec. 4, Roosevelt Rd., Taipei 10617, Taiwan; 9002090@gmail.com (S.-H.L.); twlu@ntu.edu.tw (T.-W.L.); 2Department of Orthopaedic Surgery, Chang Gung Memorial Hospital, Linkou, No. 5, Fuxing St., Guishan Dist., Taoyuan 33305, Taiwan; rick910597@gmail.com (C.-T.C.); ccc0810.chang@gmail.com (C.-H.C.); r52906154@cgmh.org.tw (C.-C.H.); b9002035@cgmh.org.tw (S.-Y.C.); 3Bone and Joint Research Center, Chang Gung Memorial Hospital, Linkou, No. 5, Fuxing St., Guishan Dist., Taoyuan 33305, Taiwan

**Keywords:** periprosthetic joint infection, total knee arthroplasty, C-reactive protein, antibiotic duration

## Abstract

Two-stage exchange arthroplasty is the standard treatment for knee periprosthetic joint infection (PJI). This study aimed to determine whether serial changes in C-reactive protein (CRP) values can predict the prognosis in patients with knee PJI. We retrospectively enrolled 101 patients with knee PJI treated with two-stage exchange arthroplasty at our institution from 2010 to 2016. We excluded patients with spacer complications and confounding factors affecting CRP levels. We tested the association between treatment outcomes and qualitative CRP patterns or quantitative CRP levels. Of the 101 patients, 24 (23.8%) had recurrent PJI and received surgical intervention after two-stage reimplantation. Patients with a fluctuating CRP pattern were more likely to receive antibiotics for a longer period (*p* < 0.001). There was greater risk of treatment failure if the CRP levels were higher when antibiotics were switched from an intravenous to oral form (*p* = 0.023). The patients who received antibiotics for longer than six weeks (*p* = 0.017) were at greater risk of treatment failure after two-stage arthroplasty. Although CRP patterns cannot predict treatment outcomes, CRP fluctuation in the interim period was associated with longer antibiotic duration, which was related to a higher treatment failure rate.

## 1. Introduction

As the number of knee arthroplasties grows, periprosthetic joint infection (PJI) is becoming one of the most devastating complications, causing 25% of revision surgeries with an incidence between 1% and 2% after primary knee arthroplasty [1,2]. The standard treatment for PJI is surgical debridement of the joint in conjunction with the implantation of an antibiotic-loaded spacer and systemic antibiotics. Although diverse treatment strategies have been introduced in the past decades, the recurrence rate of infection ranges from 11% to 28% after two-stage arthroplasty [3,4,5,6,7,8,9].

Diagnostic criteria of PJI have been established by the Musculoskeletal Infection Society (MSIS), including serum C-reactive protein (CRP). CRP is an acute phase reactant, which elevates in infectious and non-infectious conditions, including rheumatologic diseases, kidney, or liver diseases [10,11,12]. CRP plays an important role in the diagnosis of PJI, yet its trend and correlation with treatment outcome of knee PJI are not fully understood [13,14]. Many studies have shown that serum CRP is a poor predictor for persistent infection and cannot determine the optimal timing of reimplantation [3,6,15,16,17,18,19,20,21]. However, previous studies mainly focused on the CRP levels at the time of reimplantation rather than serial changes in CRP levels during the interim period of two-stage surgery, and none of them assessed the potential of using CRP levels as a guide for antibiotic regimens. We have demonstrated that CRP pattern in the interim period of two-stage surgery for hip PJI treatment is associated with treatment outcome [22]. However, the concept has not been proven in knee PJI treatment.

Systemic antibiotic treatment post-resection arthroplasty is widely used to control infection, with a recommended duration of two to six weeks using either intravenous or oral antibiotics [23]. Satisfactory outcomes have been reported with an antibiotic treatment course starting with intravenous administration to achieve desirable concentrations, followed by a switch to oral antibiotics once clinical indicators improve [19,24,25,26]. However, antibiotic regimens have been adjusted mainly depending on the duration of antimicrobial treatment, not the clinical indicators of infection status.

In the current study, we aimed to investigate the following: (1) whether the qualitative CRP patterns or quantitative CRP levels at given time points can predict the treatment outcome; and (2) whether the CRP levels can help determine the duration of antibiotic treatment. This study was necessary because serum CRP trend was an important biomarker yet a subjective observation by clinicians. We hypothesized that a gradual decline in serum CRP without fluctuation in the interim period of two-stage exchange arthroplasty was associated with higher treatment success rates in knee PJI.

## 2. Materials and Methods

### 2.1. Patient Selection

This retrospective study was approved by the Institutional Review Board of the Chang Gung Medical Foundation (#102-1846B) and was exempted from informed consent. We reviewed all patients with knee PJI who received two-stage surgery at our institution from 2010 to 2016. The patient enrollment process is outlined in Figure 1. The diagnostic criteria for knee PJI were defined according to the 2018 Musculoskeletal Infection Society (MSIS) criteria for prosthetic joint infections before resection (Table 1) [27]. All patients with knee PJI who planned to undergo two-stage revision during this time frame were included. The exclusion criteria were as follows: (1) history of inflammatory disease or autoimmune disease (n = 19); (2) concomitant infection such as urinary tract infection or pneumonia (n = 38); (3) liver cirrhosis, chronic hepatitis, or chronic kidney disease (n = 35); (4) admission to an intensive care unit (n = 6); (5) spacer complications between the two stages, including dislocation, loosening, or breaking of the spacer (n = 33); and (6) incomplete records of CRP levels between resection and reimplantation (n = 45). Among the 101 patients who met the inclusion criteria, 6 patients underwent spacer exchange due to persistent infection in the interim period, 18 patients had recurrent PJI after reimplantation, and 77 patients had their PJI successfully treated. We reviewed age, gender, body mass index (BMI), Charlson Comorbidity Index (CCI), microorganism culture results at the resection and reimplantation stages, and serial records of CRP levels. We defined a CCI of greater than 4 points as indicating a poor health condition. Acute PJI is defined as symptoms of PJI persisting for less than four weeks.

### 2.2. Treatment Protocol

All patients received two-stage surgeries, including removal of the previous prosthesis and bone cement, meticulous debridement, irrigation with 5 L of normal saline solution, and implantation of an antibiotic-loaded articulating spacer during the first stage. The two-stage exchange arthroplasties were performed by fellowship-trained joint reconstruction surgeons in the same institution with similar protocol. We obtained synovial fluid intraoperatively for routine analysis and at least 5 periprosthetic cultures, including tissue cultures and synovial fluid cultures in a blood culture bottle. The spacer was constructed with 40 g polymethylmethacrylate and empirical broad-spectrum antibiotics (4 g of vancomycin and 4 g of ceftazidime). After resection arthroplasty, patients received empirical broad-spectrum intravenous antibiotics initially, and the antibiotics regimen would be adjusted according to microbiological culture results. Serum CRP was checked in every patient immediately before resection, every week for 4 weeks after resection, and every 2 weeks until reimplantation. The antibiotics regimen was individualized and depended on the response to treatment. If clinical condition improved or CRP levels declined, we would switch to oral antibiotics or discontinue antibiotic treatment. During the interim period between the two stages, we monitored CRP levels and local infection signs via regular follow-ups. If recurrence of PJI was suspected, the patient would receive debridement and a spacer exchange for further infection control. The timing of reimplantation was decided according to a combination of patient clinical symptoms and the level of serum CRP. After two-stage arthroplasty, patients attended regular follow-ups for at least two years at our outpatient department.

### 2.3. Definition of CRP Pattern

We defined a CRP trend as a serial change in CRP level and classified trends into five categories according to whether the CRP value dropped to a normal range and the timing of the CRP level normalization (Figure 2 and Figure 3). We used a threshold of 10 mg/L for normal CRP levels and the third week after resection arthroplasty as the defining point for an “ideal” drop or delayed drop in CRP level. The five patterns included: (1) CRP level dropped to the normal range within three weeks; (2) CRP level dropped to the normal range after three weeks; (3) CRP level was never above the normal range; (4) CRP level fluctuated but had been normal at any time point; and (5) CRP level fluctuated and never dropped within the normal range. We also evaluated CRP values at the following given time points: pre-resection, at the discontinuation of intravenous antibiotics, at the discontinuation of all antibiotics, and pre-reimplantation. CRP changes between the four time points were investigated and correlated with treatment outcome.

### 2.4. Definition of Raecurrent PJI and Treatment Failure

Recurrent or persistent PJI was defined as reinfection after two-stage exchange arthroplasty according to the 2018 MSIS criteria: persistent presentation of sinus tract communication with the joint space and presence of the same microorganism in two intraoperative cultures [27]. The definition of treatment failure was based on the modified Delphi-based criteria: (1) the presence of a sinus tract infection, an unhealed wound or weeping, or infection recurrence caused by the same organism strain; (2) any unplanned surgical intervention, except reimplantation surgery; or (3) occurrence of PJI-related mortality [28].

### 2.5. Statistical Analysis

The primary outcome measure was cure of PJI. We analyzed the correlation between the treatment outcome and factors including patient demographics, microbiologic result, CRP patterns, CRP values at four time points, antibiotic regimens (intravenous and oral), and eventual antibiotic washout before reimplantation. Mann–Whitney *U* tests and Fisher’s exact tests were used for numerical and categorical comparisons, respectively. Multivariate logistic regression was used to evaluate the correlation of CRP pattern and treatment outcome. The relationship between either the CRP pattern or type of pathogen and the length of the antibiotic duration was analyzed with one-way analysis of variance. Statistical analysis was conducted with IBM SPSS Statistics (version 22.0, IBM, New York, NY, USA). All tests were two-sided, with *p* < 0.05 indicating statistical significance.

## 3. Results

In total, 101 patients with knee PJI were enrolled in this study, 43 of whom were men (43%) with a mean age of 70 years old (ranging from 30 to 80 years old). The mean body weight was 68.76 kg (ranging from 43 to 115.9 kg), and the mean BMI was 28.1 kg/m^2^ (ranging from 17.2 to 45.2 kg/m^2^). The patients’ characteristics, microbiology, CRP profiles, and antibiotic duration are listed in Table 2. Sinus tract was present in 24 patients (24%), and multiple sets of positive culture results were noted in 43 patients (43%). Aerobic and anaerobic cultures were obtained from multiple intraoperative samples, and the isolated pathogens were as follows: 36 methicillin-sensitive *Staphylococcus aureus*, 10 methicillin-resistant *S. aureus* (MRSA), 6 other Gram-positive bacteria (including two *Staphylococcus capitis,* two *Streptococcus* species, and two *Enterococcus* species), 7 Gram-negative bacteria (including two *Pseudomonas* species, 3 *Klebsiella* species, 2 *Serratia marcescens*), 4 fungal infections (all *Candida* species), and 4 mixed florae (including one patient with *Streptococcus viridans* and *Peptostreptococcus micros*, one with *Staphylococcus lugdunensis* and *Penicillium* species, one with vancomycin-resistant *Enterococcus* and *Klebsiella pneumonia*, and one with MRSA and *Candida* species). No pathogens were isolated in 34 patients. We found that heavier body weight may increase the risk of treatment failure (*p* = 0.006). However, the microbiology results could not predict the outcome of the two-stage surgery. Overall, infection was successfully eradicated in 77 of 101 patients (77%). The average interim period between the two stages was 14.6 weeks (ranging from 6 to 75 weeks). Six patients received spacer exchange during the interim period; one received spacer exchange due to mycobacterial infection, whereas the others received additional surgery due to signs of persistent infection. The infection was controlled in three of the six patients. Neither the length of the interim period nor receiving interim surgery indicated the outcomes of the two-stage surgery.

Table 2 shows the CRP patterns and CRP values at given time points. Forty-one patients (41%) had their CRP levels drop back to normal within three weeks (type 1), and 42 patients (42%) saw their CRP levels fall within the normal range after three weeks (type 2). Seven patients (7%) showed no elevated of CRP levels (type 3). The remaining seven (7%) and four patients (4%) had fluctuating CRP levels with (type 4) or without (type 5) recorded normal CRP values, respectively. There was no statistically significant correlation between CRP pattern and PJI treatment outcome (*p* = 0.186). However, there was higher possibility of treatment failure if the patient had type 5 CRP pattern (50.0%) than type 1 (19.5%) or type 2 (21.4%). In addition, the changes in CRP values between given time points showed no significant difference among the treatment success and treatment failure groups. It is worth noting that, when intravenous (IV) antibiotics were discontinued, the CRP levels were significantly higher in the treatment failure group (27.1 mg/L vs. 13.1 mg/L; *p* = 0.023). The cutoff value when IV antibiotics were discontinued was 29.15 mg/L (Sensitivity: 0.38; Specificity: 0.94; PPV: 64%; NPV: 85%).

The average total antibiotic administration duration was 5.5 weeks (ranging from 0 to 27.5 weeks), and the average length of IV antibiotic use was 2.3 weeks (ranging from 0 to 7.5 weeks). Antibiotic regimens longer than six weeks (odds ratio: 3.086; 95% confidence interval: 1.194–7.979; *p* = 0.017) and a lack of antibiotic washout before reimplantation surgery (odds ratio: 3.34; 95% confidence interval: 0.09–1.003; *p* = 0.042) were associated with treatment failure (Table 2).

In addition, the patients with type 4 (*p* = 0.011) and type 5 CRP patterns (*p* = 0.026) received longer antibiotic treatment during the interim period compared to patients with type 1 (Table 3), which indicates that we were prone to using antibiotics for a longer time with patients with fluctuating CRP patterns due to the fact that it might be related to a poor response to PJI management. The initial pathogen was not associated with the length of the antibiotic regimen for our patients.

## 4. Discussion

In the present study, we discovered there was no statistically significant correlation between CRP pattern and treatment outcome of knee PJI. However, the data showed a higher treatment failure rate of patients with type 5 CRP pattern compared with type 1 or type 2. This finding confirmed our impression of CRP interpretation clinically that a gradual decline of CRP correlated to a higher success rate than a fluctuating CRP pattern in terms of PJI treatment.

Several studies have described CRP trends as a critical parameter for assessing PJI treatment response after the first-stage resection in a two-stage exchange arthroplasty [3,5,15,16,21,29]. Previous authors have failed to set a threshold for CRP levels to guide the timing of reimplantation [3,6,15,16,17,18,19,20,21]. We found that neither the qualitative CRP patterns nor the quantitative CRP values at given time points during the interim period could predict the outcome of two-stage exchange arthroplasty. In the previous report, we found poor serial CRP response to treatment, namely type 4 and 5 CRP patterns, was an indicator for poor prognosis for patients with hip PJI undergoing two-stage exchange arthroplasty [22]. Nevertheless, the phenomenon was not evident in the present study, presumably because of a relatively small sample size to reach statistical significance. We did, however, find that type 4 and 5 CRP patterns were associated with longer total antibiotic duration (Table 3), which, in turn, was related to higher failure rate if total antibiotic duration was longer than 6 weeks (Table 2).

Serial CRP level measurement is widely used to monitor the clinical responses of patients with infectious diseases, but few authors have studied the diagnostic value of serial CRP level measurements. Bejon et al. examined 3732 measurements from 260 patients and showed AUC values ranging from 0.55 to 0.65, which indicates that serial CRP level changes cannot accurately predict outcomes [20]. Ghani et al. showed that the CRP preoperative values and at weeks 1, 3, and 6 post-operation were poor indicators of outcomes not only in patients undergoing two-stage surgery but also in patients undergoing one-stage surgery with debridement, implant retention and subsequent antibiotic therapy [30]. Stambough et al. assessed the change in CRP levels between pre-resection and six weeks after intravenous antibiotic treatment, but the change could not accurately determine the timing of reimplantation [21]. We categorized the patterns of serial CRP values into five types to represent different responses to treatment, but these patterns did not predict outcomes after two-stage surgery. However, patterns 4 and 5 (patterns with fluctuating CRP values) were associated with longer antibiotic regimens, which indicates that the patients with poor CRP patterns may need longer antibiotic regimens to control infection due to unsatisfactory management responses.

The antibiotic-loaded spacer and antimicrobial treatment were the main treatments for this two-stage surgery. The 2018 international consensus meeting recommends antibiotic therapy for two to six weeks, either intravenously or orally [23]. Many studies have reported satisfactory outcomes for eradication of infection through even shorter intravenous regimens [4,7,8,24,31,32,33]. Prolonged intravenous antibiotic use may cause more complications, such as catheter-related infections, which can sometimes result in longer admission periods and greater medical expenditures [31,34,35,36,37,38]. Bernard et al. indicated that the length of intravenous antibiotic administration did not change the cure rate of PJI, with a maximum of six weeks of antibiotic therapy [39]. Sandiford et al. showed that there was no statistically significant difference between the duration of intravenous antibiotic therapy and the cure rate of PJI [40]. At our institution, we individualized the patients’ antibiotic regimens based on clinical symptoms and laboratory data rather than using a fixed duration for all patients. To our knowledge, there has been no literature discussing the timing for antibiotic shift from IV to oral form. In our study, we found that CRP values when discontinuation of IV antibiotics were related to the outcome of the two-stage surgery. Higher CRP values were related to higher risk of treatment failure. Although the diagnostic power of CRP level during the discontinuation of intravenous antibiotics was not satisfactory, we did provide a potential reference for clinicians to determine the following treatment. If the CRP value is higher than 29.15 mg/dL, it is better to continue IV antibiotics or consider debridement for better infection control.

We also found that heavier body weight, an antibiotic regimen lasting longer than six weeks, and a lack of drug holiday before reimplantation were risk factors for treatment failure after reimplantation surgery. Prolonged antibiotic regimens were related to poor responses to treatment. Although drug holidays are used to allow for emergence of residual infection, Bejon et al. pointed out that the drug holiday was not essential because most of the re-debridement happened when the patients were under antibiotic treatment, and the rate of positive microbiological results during reimplantation were similar between patients with or without drug holidays (13% vs. 16%) [41]. However, a higher risk of treatment failure was seen in our patients without a drug holiday. Elevated CRP values were recorded in only one of 13 patients without drug holiday. The finding was reasonable for this retrospective study because for patients with poor clinical or CRP patterns, we were more prone to keep antibiotics in the interim period.

Based on our findings, clinicians could monitor the serial changes of serum CRP values according to the CRP patterns we proposed, rather than a time-point CRP value, which is easily affected by various conditions. Patients with type 4 and type 5 CRP tended to receive longer antibiotic treatment, yet still higher treatment failure rate. As a result, a different treatment strategy, such as an aggressive and early debridement in the interim period, might be considered.

Our study has several limitations. First, the case number of patients with the type 4 or type 5 patterns was small. This limits the power of this study and increases the difficulty in analyzing the relationship between CRP patterns and outcome. Second, the length of the antibiotic duration was determined by evaluation of clinical condition and laboratory data by different physicians. Although some consensus of clinical practice was established, there remained different views when evaluating the treatment response of our patients. Finally, CRP is a sensitive acute phase reactant, which fluctuates in many clinical scenarios.

We strove to eliminate confounding factors of CRP, such as the same treatment protocol, surgical approach, timing of CRP test, and excluding patients with pre-existing kidney or liver diseases, autoimmune diseases, concomitant infections, and spacer complications [11,22]. However, there were still various uncontrollable factors and systemic conditions that could affect CRP values. This study aimed to provide evidence to verify our long-established practice of using CRP trends as a guide for clinical decisions. Further prospective controlled studies will be needed to better understand CRP trends and their diagnostic power.

## 5. Conclusions

There was a trend that fluctuating CRP patterns were associated with higher treatment failure rates in patients with knee PJI undergoing two-stage exchange arthroplasty, although not statistically significant. CRP fluctuation in the interim period was associated with longer antibiotic duration, which was related to higher treatment failure rate. Further prospective studies with larger case numbers are necessary to determine the optimal duration of antibiotic treatment.

## Figures and Tables

**Figure 1 diagnostics-12-01030-f001:**
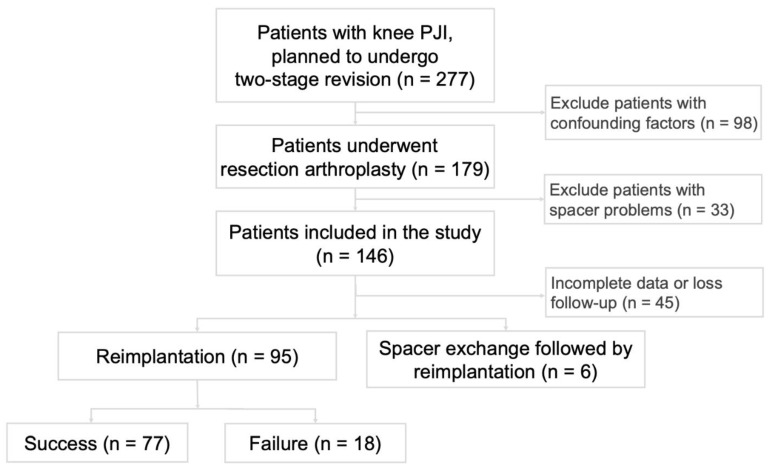
Patient enrollment algorithm. Three out of six patients who had spacer exchange did not have infection recurrence, but still considered treatment failure according to Delphi criteria.

**Figure 2 diagnostics-12-01030-f002:**
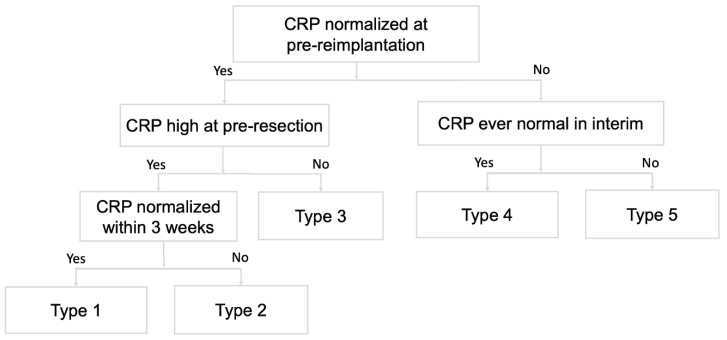
Qualitative CRP trends. The CRP trends during the interim period are categorized into 5 types according to the CRP levels at pre-reimplantation, pre-resection, and whether CRP is normalized within 3 weeks after resection.

**Figure 3 diagnostics-12-01030-f003:**
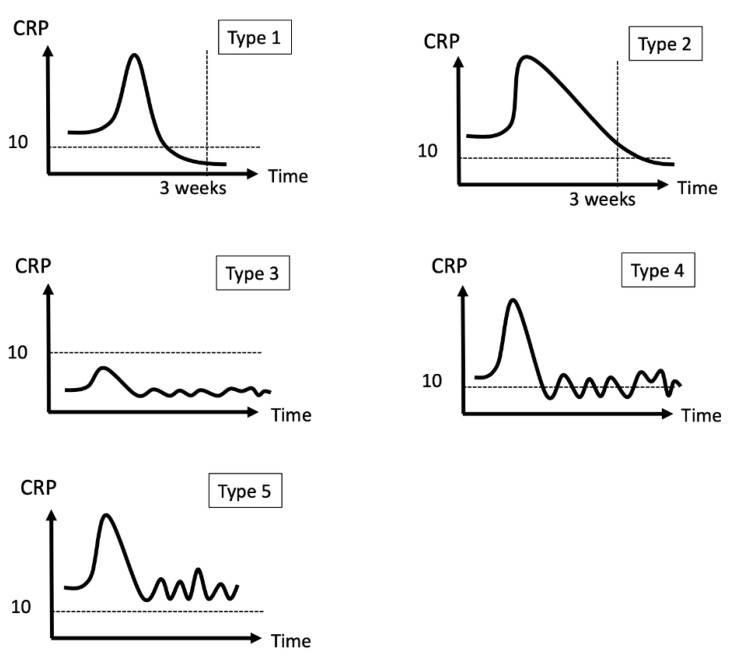
Examples of five CRP patterns. CRP usually elevated right after resection.

**Table 1 diagnostics-12-01030-t001:** Definition of periprosthetic joint infection. It was a scoring-based definition adopted by the Musculoskeletal Infection Society [27].

Major Criteria (at Least One of the Following)			Decision	
Two Positive Cultures of the Same Organism			Infected	
Sinus tract with evidence of communication to the joint or visualization of the prosthesis				
**Minor criteria**			Score	Decision
**Pre-OP diagnosis**	Serum	Elevated CRP or D-Dimer	2	≥6 Infected
		Elevated ESR	1	
	Synovial	Elevated synovial WBC or LE	3	2–5 Possibly infected
		Positive alpha-defensin	3	
		Elevated synovial PMN (%)	2	0–1 Not infected
		Elevated synovial CRP	1	
**Inconclusive Pre-OP score, or dry tap**			Score	Decision
**Intra-OP diagnosis**	Pre-OP score		-	
	Positive histology		3	≥6 Infected
	Positive purulence		3	4–5 Inconclusive
	Single positive culture		2	≤3 Not infected

**CRP:** C-reactive protein, **ESR:** erythrocyte sedimentation rate, **WBC:** white blood cell, **LE:** leukocyte esterase, **PMN:** polymorphonuclear leukocyte, **OP:** operative.

**Table 2 diagnostics-12-01030-t002:** Association of treatment outcome and variables. Treatment outcome was defined by modified Delphi criteria.

	Variable	Success (n = 77)	Failure (n = 24)	*p* Value
Patient characteristics	Age (years)	70.8	67.2	0.164
	Male	27 (35%)	16 (67%)	0.006 *
	Body weight	66.6	75.7	0.006 *
	BMI	28	30.1	0.166
	CCI			
	4<	26 (34%)	11 (46%)	0.284
	≧4	51 (66%)	13 (54%)	
	Presence of sinus tract	19 (25%)	5 (21%)	0.699
	Interim period (weeks)	13.9	16.6	0.339
Microbiology	Culture-negative	25 (33%)	9 (38%)	0.883
	MSSA	28 (36%)	8 (33%)	
	MRSA	7 (9%)	3 (13%)	
	Other G (+)	6 (8%)	0	
	G (-)	5 (6%)	2 (8%)	
	Fungus	3 (4%)	1 (4%)	
	Polymicrobial	3 (4%)	1 (4%)	
CRP pattern	Type 1	33	8	0.186
	Type 2	33	9	
	Type 3	4	3	
	Type 4	5	2	
	Type 5	2	2	
CRP characteristics	Pre-resection CRP	68.5 (1.2–353.4)	68 (5.7–272.8)	0.977
	CRP when DC IV Abx	13.1 (0.8–46.65)	27.1 (1–122)	<0.001 *
	∆ Preresection-DC IV Abx	−13%	73%	0.190
	CRP when DC all Abx	8 (0.5–45.6)	9.2 (1–35.4)	0.564
	∆ Preresection-DC all Abx	−62%	−77%	0.313
	Pre-reimplantation CRP	6.1 (0.45–28.9)	8 (0.9–37.8)	0.249
	∆ Preresection-pre-reimplantation	−66%	−71%	0.745
Antibiotics	IV Abx duration			
	Mean(days)	15	19	0.212
	≦14 days	46 (60%)	11 (46%)	0.23
	>14 days	31 (40%)	13 (54%)	
	Total Abx duration			
	Mean (weeks)	5.2	6.4	0.195
	≦6 weeks	50 (65%)	9 (38%)	0.017 *
	>6 weeks	27 (35%)	15 (62%)	
	Abx after reimplantation	18 (23%)	7 (29%)	0.566
	Abx drug holiday	70 (91%)	18 (75%)	0.042 *

**Abx**: antibiotic, **BMI:** body mass index, **CCI:** Charlson, comorbidity index, **DC**: discontinue, **IV**: intravenous, **MRSA:** methicillin-resistant *Staphylococcus aureus*, **MSSA**: methicillin-sensitive *Staphylococcus aureus.* * *p* < 0.05.

**Table 3 diagnostics-12-01030-t003:** Antibiotics duration. CRP type 4 and type 5 were associated with longer total antibiotic use.

	Mean (Weeks)	95% CI	*p* Value
CRP pattern			<0.001 *
1	4.3	3.4–5.3	
2	5.5	4.6–6.4	
3	4.8	1.5–8.0	
4	9.9	2.3–17.6	
5	10.9	1.2–20.6	
Microbiology			0.089
Culture-negative	4.1	3.2–5.1	
MSSA	5.6	3.2–7.9	
MRSA	6.1	4.4–7.7	
other G (+)	4.6	3.6–5.5	
G (-)	7.3	3.8–10.8	
Fungus	6.0	0.2–11.8	
Polymicrobial	9.6	−0.4–19.7	

**CI**: confidence interval, **MRSA**: methicillin-resistant *Staphylococcus aureus*, **MSSA:** methicillin-sensitive *Staphylococcus aureus.* * *p* < 0.05.

## Data Availability

The datasets generated during the current study are available from the corresponding author upon reasonable request.

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
