# Peer review of "Do Serum C-Reactive Protein Trends Predict Treatment Outcome in Patients with Knee Periprosthetic Joint Infection Undergoing Two-Stage Exchange Arthroplasty?"

_diagnostics, 2022, doi:10.3390/diagnostics12051030_

Round 1

Reviewer 1 Report

This is an interesting study on the relationship between CRP fluctuations and success of two-stage exchange arthroplasty for PJI. The manuscript is well written and the hypothesis of the authors is interesting, with a few points to be addressed before considering publication.

Some comments:

  • It is not clear whether the patients included in the study underwent CPR evaluation at the same timepoints and how many CPR datapoints are available for each patient. Differences in dosage timing may contribute to further fluctuations and inconsistency that can alter the results. Authors should deeply discuss this point.
  • Line 41: Please add "the implantation of" before "an antibiotic-loaded".
  • Information about the surgeon(s) performing the operations should be provided. Was it (or were they) the same for each stage? 
  • Lines 134-135: please change "Initially, the primary measurable outcome was cure of PJI. We analyzed the correlation between the treatment outcome and factors" to "The primary outcome measure was cure of PJI. We analyzed the correlation between the treatment outcome and other factors".
  • Line 137: Please change "incorporation of drug holiday or not" to "eventual antibiotic washout".
  • Table 1 caption: Please replace "methillin" with "methillicin" in MSSA and MRSA extended forms and reorganize the abbreviations all in the same line and in alphabetical order. Same in Table 2 caption.
  • Lines 152-156: All isolated microorganism should be detailedly mentioned.
  • Line 170: Please change "drop" to "dropped".
  • Line 172: Please change "had no elevated" to "showed no elevation".
  • Line 184: Please replace "drug holiday" with "antibiotic washout".
  • Line 187: Please remove "the".
  • Line 189: Please change "longer" to "for a longer time" and remove "the".
  • Line 215: Please replace "value pre-operation" with "preoperative values".
  • Line 217: Please replace "the debridement, antibiotics and implant retention and one-stage surgery" with "one-stage surgery with debridement, implant retention and subsequent antibiotic therapy".
  • Line 218: Please change "used" to "assessed".
  • Another major limitation is associated with the nature of CRP itself, whose fluctuations may be influenced by several factors including patient variability, surgical approach etc. This should be discussed as well.

Author Response

Dear Reviewer,

Thank you for your precious time to review the manuscript, and providing valuable suggestions to the article. The authors’ responses to the comments are as follows:

  • It is not clear whether the patients included in the study underwent CPR evaluation at the same timepoints and how many CPR datapoints are available for each patient. Differences in dosage timing may contribute to further fluctuations and inconsistency that can alter the results. Authors should deeply discuss this point.

Reply: Thank you for raising the critical point. Serum CRP was checked in every patient right before resection, every one week for 4 weeks after resection, and every 2 weeks until reimplantation. The average interval between two stages were 12 weeks, that yielded 8 CRP values in the interim period. The timepoints for CRP is added to the “treatment protocol” section.

  • Line 41: Please add "the implantation of" before "an antibiotic-loaded".

Reply: It is rephrased accordingly.

  • Information about the surgeon(s) performing the operations should be provided. Was it (or were they) the same for each stage? 

Reply: The two-stage exchange arthroplasties were performed by fellowship-trained joint reconstruction surgeons in the same institution with similar protocol. I have described in more detail in “treatment protocol” section.

  • Lines 134-135: please change "Initially, the primary measurable outcome was cure of PJI. We analyzed the correlation between the treatment outcome and factors" to "The primary outcome measure was cure of PJI. We analyzed the correlation between the treatment outcome and other factors".

Reply: It is rephrased accordingly.

  • Line 137: Please change "incorporation of drug holiday or not" to "eventual antibiotic washout".

Reply: It is rephrased accordingly.

  • Table 1 caption: Please replace "methillin" with "methillicin" in MSSA and MRSA extended forms and reorganize the abbreviations all in the same line and in alphabetical order. Same in Table 2 caption.

They are corrected.

  • Lines 152-156: All isolated microorganism should be detailedly mentioned.

Reply: They are described in more detail.

  • Line 170: Please change "drop" to "dropped".
  • Line 172: Please change "had no elevated" to "showed no elevation".
  • Line 184: Please replace "drug holiday" with "antibiotic washout".
  • Line 187: Please remove "the".
  • Line 189: Please change "longer" to "for a longer time" and remove "the".
  • Line 215: Please replace "value pre-operation" with "preoperative values".
  • Line 217: Please replace "the debridement, antibiotics and implant retention and one-stage surgery" with "one-stage surgery with debridement, implant retention and subsequent antibiotic therapy".
  • Line 218: Please change "used" to "assessed".

Reply: Thank you for the above-mentioned suggestions. They are rephrased accordingly.

  • Another major limitation is associated with the nature of CRP itself, whose fluctuations may be influenced by several factors including patient variability, surgical approach etc. This should be discussed as well.

Reply: Thank you for raising the point. CRP truly is a sensitive biomarker that is easily affected by various clinical conditions. So we tried to exclude confounding factors of CRP to avoid these situations. I have elaborated more in the “limitation” section.

Reviewer 2 Report

Dear Authors,

Thank you very much for your contribution in Diagnostics. The article is very interesting and is worthy of publication. However, there are some issues that should be addressed:

Please use third voice and past tense

INTRODUCTION

Please add a hypothesis

Please try to clearly state in a sentence why this study is necessary

METHODS

Please add a table describing the 2018 musculoskeletal infection society 73 (MSIS) criteria for prosthetic joint infections

Please refer to the guidelines for the study (CONSORT, STROBE). All clinical studies should be conducted according current guidelines

State all the ethical precautions in the methods: ethic vote, declaration of Helsinki, patients consensus

The statistical analyses have not been sufficiently described. Please describe carefully each analysis, software, threshold, how you conduct the analysis and data interpretation. This paragraph is one of the most important and should be point per point clearly reported

RESULTS

Please add a full description and flow chart diagram of the recruitment process which lead to the studies cohort of patients

DISCUSSION

Please start the discussion describing your findings

Please add a paragraph comparing your results with those of previous studies

Please expand the limitations. These are not enough

Please add the clinical relevance of your study and some practical application of your findings

Author Response

Dear Reviewer,

Thank you for your precious time to review the manuscript, and providing valuable suggestions to the article. The authors’ responses to the comments are as follows:

  • Please use third voice and past tense

Reply: Thank you. I have modified the article to past tense.

  • INTRODUCTION
  • Please add a hypothesis. Please try to clearly state in a sentence why this study is necessary

Reply: Thank you. A hypothesis and necessity of the study are added to the end of “Introduction” section.

  • METHODS
  • Please add a table describing the 2018 musculoskeletal infection society 73 (MSIS) criteria for prosthetic joint infections.

Reply: Thank you for the suggestion. A table (table 1) has been added to describe MSIS criteria.

  • Please refer to the guidelines for the study (CONSORT, STROBE). All clinical studies should be conducted according current guidelines

Reply: Thank you for the suggestion. In this retrospective case-control study, we complied with STROBE guidelines.

  • State all the ethical precautions in the methods: ethic vote, declaration of Helsinki, patients consensus

Reply: I added the ethical precaution in the “Materials and Methods” section. Due to the retrospective nature of the study and the routine medical practice, we did not pose harm to the patients.

  • The statistical analyses have not been sufficiently described. Please describe carefully each analysis, software, threshold, how you conduct the analysis and data interpretation. This paragraph is one of the most important and should be point per point clearly reported.

Reply: Thank you for the suggestion. I have described more clearly regarding the statistical analysis.

  • RESULTS
  • Please add a full description and flow chart diagram of the recruitment process which lead to the studies cohort of patients

Reply: Thank you for the suggestion. Figure 1 is a flow chart diagram of the recruitment process. I have also described the recruitment process in the “patient selection” section.

  • DISCUSSION
  • Please start the discussion describing your findings

Reply: Thank you for the suggestion. They are added to the beginning of discussion.

  • Please add a paragraph comparing your results with those of previous studies

RepIy: I have modified accordingly.

  • Please expand the limitations. These are not enough

Reply: I have discussed more in “limitations” section.

  • Please add the clinical relevance of your study and some practical application of your findings

Reply: Thank you. I have added a paragraph of clinical relevance of the study in front of “limitation” section.

Round 2

Reviewer 1 Report

The authors have significantly improved their manuscript according to reviewers' suggestion.

Other minor changes needed:

  • Please mark all the changes that have been performed during the review process to allow a better tracking.
  • Table 1: Please cite the corresponding reference in the Table caption and revise the Table to enhance readability (correct typos, blank spaces and sentences in the left column).
  • Line 102, Treatment Protocol paragraph: please clearly state who performed the surgeries as indicated in the response to reviewers file.
  • Lines 176-177: Detailed information about isolated fungi and mixed flora should be stated as well.

Author Response

Dear Reviewer,

Thank you again for reviewing the manuscript, and providing valuable suggestions to the article. The authors’ responses to the comments are as follows:

  • Please mark all the changes that have been performed during the review process to allow a better tracking.

Reply: I have uploaded a revised version with track changes.

  • Table 1: Please cite the corresponding reference in the Table caption and revise the Table to enhance readability (correct typos, blank spaces and sentences in the left column).

Reply: Thank you. I have corrected them.

  • Line 102, Treatment Protocol paragraph: please clearly state who performed the surgeries as indicated in the response to reviewers file.

Reply: Thank you for the comment. It has been modified accordingly.

  • Lines 176-177: Detailed information about isolated fungi and mixed flora should be stated as well.

Reply: Thank you. The information has been added to the manuscript.

Reviewer 2 Report

no more comments. thank you for the revision

Author Response

Dear Reviewer, 

Thank you for your precious time to review the manuscript. 

Sincerely,

Yu-Chih Lin.